# Language Representation Projection: Can We Transfer Factual Knowledge across Languages in Multilingual Language Models?

**Shaoyang Xu**[1],  **Junzhuo Li**[1] and **Deyi Xiong**[21*]

[1]School of New Media and Communication, Tianjin University, Tianjin, China
[2]College of Intelligence and Computing, Tianjin University, Tianjin, China
{syxu, jzli, dyxiong}@tju.edu.cn

## Abstract

Multilingual pretrained language models serve as repositories of multilingual factual knowledge. Nevertheless, a substantial performance gap of factual knowledge probing exists between high-resource languages and low-resource languages, suggesting limited implicit factual knowledge transfer across languages in multilingual pretrained language models. This paper investigates the feasibility of explicitly transferring relatively rich factual knowledge from English to non-English languages. To accomplish this, we propose two parameter-free **L**anguage **R**epresentation **P**rojection modules (LRP2). The first module converts non-English representations into English-like equivalents, while the second module reverts English-like representations back into representations of the corresponding non-English language. Experimental results on the mLAMA dataset demonstrate that LRP2 significantly improves factual knowledge retrieval accuracy and facilitates knowledge transferability across diverse non-English languages. We further investigate the working mechanism of LRP2 from the perspectives of representation space and cross-lingual knowledge neuron.

## 1  Introduction

Previous studies demonstrate that a language model is a knowledge base that can recall factual knowledge without additional fine-tuning (Petroni et al., 2019; Jiang et al., 2020b). This task of factual knowledge probing, aiming to examine what factual knowledge language models capture during the pre-training phase, can be extended to multiple languages in multilingual pretrained language models, e.g., mBERT (Devlin et al., 2019), XLM (Conneau and Lample, 2019), mT5 (Xue et al., 2021), XGLM (Lin et al., 2022) and BLOOM (Scao et al., 2022). Although multilingual pretrained models serve as repositories of multilingual factual knowledge, a factual knowledge gap exists

---
* Corresponding author.

between English and other languages in terms of the amount of factual knowledge captured for each language (Kassner et al., 2021; Jiang et al., 2020a).

Many works on cross-lingual transfer (Conneau et al., 2020; Chi et al., 2021; Wu et al., 2022; Yang et al., 2022) validate the effectiveness of cross-lingual alignment of representation spaces in facilitating cross-lingual knowledge transfer. These studies primarily evaluate their methods on specific downstream tasks, including natural language inference (Conneau et al., 2018), sentence retrieval (Artetxe and Schwenk, 2019), question answering (Lewis et al., 2020) and text generation (Wu et al., 2022), etc.

Different from such studies, we focus on the task of factual knowledge probing in multilingual pretrained language models and attempt to answer a question in this paper: *Can cross-lingual alignment of representation spaces enable factual knowledge transfer across languages?* In particular, we explore the feasibility of transferring factual knowledge from English to non-English languages.

To answer this question, we propose LRP2, which incorporates two parameter-free **L**anguage **R**epresentation **P**rojection modules into multilingual pretrained models: a language-independent representation projection module that projects representations of non-English languages into English-like representations and a language-specific representation projection module that maps the English-like representations back to representations of individual non-English languages. These two modules, as depicted in Figure 1, locate at different layers of Transformer.

Experiments on mLAMA (Kassner et al., 2021) suggest that LRP2 improves factual knowledge retrieval accuracy and facilitates knowledge transfer across diverse languages. We further conduct in-depth analysis to investigate the varying degrees of representation alignment required by different non-English languages, as well as the transferabil-

ity of different types of factual knowledge. Delving into the working mechanism of LRP2, we identify cross-lingual knowledge neurons in multilingual pretrained language models.

Our contributions are summarized as follows.

- We propose a parameter-free framework LRP2 that enhances factual knowledge retrieval accuracy and cross-lingual factual knowledge transfer.

- We reveal that LRP2 poses an impact on the alignment of representation spaces and enhances the overlap of knowledge neurons across languages.

- We discover that cross-lingual knowledge neurons exist in multilingual language models.

## 2 Multilingual Factual Knowledge Probing

In the multilingual factual knowledge probing task, multilingual pretrained language models take language-specific fill-in-the-blank queries as input, such as "The capital of England is [MASK]" in English, or the corresponding Chinese question "英国的首都是[MASK]". As a knowledge base, the probed pretrained language model initially encodes the input query, then retrieves its parameterized memory and ultimately predicts an answer with a probability distribution over the vocabulary.

The success of factual knowledge transfer across languages relies on a language-independent representation space for different languages to trigger similar memories within the probed multilingual pretrained model and language-specific representations to allow the model to predict tokens in the corresponding language.

## 3 LRP2

The primary objective of LRP2 is to bridge the gap of factual knowledge probing between English and non-English languages by aligning their representation spaces.

Libovický et al. (2020) demonstrate that it is possible to induce language-neutral representations for a given language, by subtracting its corresponding language vector. The proposed LRP2 draws inspiration from this work and initiates its process by computing a set of language vectors $\mathcal{V}_l$ for each language $l$. Specifically, for language $l$, we feed a set of its sentences into the multilingual

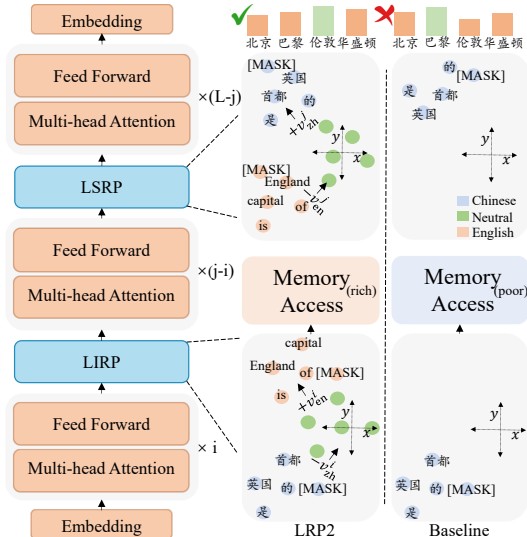

Figure 1: The diagram of the proposed LRP2 that inserts two language representation projection modules as additional layers into the multilingual pretrained language model. The input question is "英国的首都是[MASK]". $\boldsymbol{v}_{zh}^i, \boldsymbol{v}_{en}^i, \boldsymbol{v}_{zh}^j$ and $\boldsymbol{v}_{en}^j$ represent language vectors obtained for Chinese and English from the $i$-th and $j$-th layer of the multilingual pretrained language model in advance. We use Chinese to showcase our framework and our method is applicable to other languages in the same way. For simplicity, we ignore other sublayers in Transformer in the diagram. Note that our method is based on the core assumption that the representation spaces of two languages can be transferred through a Euclidean distance mapping. This form of straightforward mapping is relatively coarse, incapable of achieving the level of precise semantic transfer depicted in the figure, which is presented for the sake of illustration but may appear somewhat overly idealized.

pretrained language model to be probed. From the $i$-th layer of the model, we gather sentence-level vectors through mean-pooling over the representations of all tokens in the corresponding sentence. We then further average these sentence vectors, obtaining $\boldsymbol{v}_l^i \in \mathbb{R}^n$, where $n$ is the hidden dimension of the model. In this way, we collect a set of vectors $\mathcal{V}_l = [\boldsymbol{v}_l^1, \boldsymbol{v}_l^2, ..., \boldsymbol{v}_l^L]$, where $L$ denotes the number of layers of the model. These language vectors serve as the basis for language representation projection within the proposed LRP2 framework.

As illustrated in Figure 1, LRP2 incorporates two language representation projection modules into the probed multilingual pretrained language model, which are referred to as the **L**anguage-**I**ndependent **R**epresentation **P**rojection (LIRP) module and the **L**anguage-**S**pecific **R**epresentation **P**rojection (LSRP) module, respectively. These two modules

| Model | English (Source) | Language Family | | Language Resource | | | Avg |
|---|---|---|---|---|---|---|---|
| | | Indo-European | non-Indo-European | High | Medium | Low | |
| | | *Retrieval Accuracy* | | | | | |
| mBERT | 35.2 | 20.9 | 18.4 | 23.4 | 22.2 | 17.4 | 20.0 |
| mBERT (LRP2) | 35.2 | 21.2 | 19.4 | 24.1 | 23.0 | 17.7 | 20.6 |
| BLOOM | 35.1 | 17.8 | 18.4 | 21.7 | 17.2 | 16.1 | 18.0 |
| BLOOM (LRP2) | 35.1 | 21.3 | 22.4 | 25.8 | 21.2 | 19.3 | 21.7 |
| | | *English-centric Cross-lingual Transferability* | | | | | |
| mBERT | 1 | 37.0 | 31.8 | 41.6 | 37.7 | 30.5 | 35.2 |
| mBERT (LRP2) | 1 | 37.9 | 33.1 | 43.1 | 38.5 | 31.5 | 36.3 |
| BLOOM | 1 | 20.4 | 20.3 | 25.7 | 19.3 | 17.6 | 20.4 |
| BLOOM (LRP2) | 1 | 24.5 | 24.7 | 30.3 | 24.0 | 21.4 | 24.6 |

Table 1: Evaluation results on mLAMA. We report factual knowledge retrieval accuracy and English-centric cross-lingual transferability. We list average results for Indo-European, non-Indo-European, high-resource, medium-resource, low-resource and all non-English languages. We measure the amount of language resource based on the number of Wikipedia articles for each language.

are inserted into the model as two additional layers. Representations of a non-English language with limited information are projected to the English representation space by LIRP, which enables the non-English language to access relatively rich memory encoded in the parameters of the model, in the form of English-like representations. The accessed memory is then projected back to the non-English language by LSRP so that answers in the corresponding non-English language can be yielded.

Specifically, given an input query in a non-English language $l$, the LIRP first projects the contextual representations from the $i$-th layer of the model into English-like representations, which can be formulated as follows:

$$\hat{h}_l^i = h_l^i - v_l^i + v_{en}^i \quad (1 \le i < L) \qquad (1)$$

where $h_l^i$ represent the $i$-th layer hidden states of the input query in language $l$. $v_l^i$ and $v_{en}^i$ denote the language vectors of the $i$-th layer for non-English language $l$ and English respectively. By performing this projection, the representations of non-English language $l$ are mapped into the English space and subsequently fed to the succeeding layers.

As mentioned in Section 2, in the multilingual factual knowledge probing task, it is essential for the multilingual pretrained language model to yield answers in the corresponding language. To recover the language-specific information of the input language, we insert the LSRP into the $j$-th layer of the model. The back-projection to the input language is formulated as:

$$\hat{h}_l^j = h_l^j - v_{en}^j + v_l^j \quad (i < j \le L) \qquad (2)$$

where $h_l^j$ represent the $j$-th layer hidden states of the input query in language $l$. $h_l^j$ are English-like representations because of the first projection. They are transformed back into the language $l$'s representation space, resulting in $\hat{h}_l^j$. These language-specific representations are further fed to the succeeding layers of the model.

## 4 Experiments

We conducted extensive experiments to examine the effectiveness of the proposed LRP2 framework in factual knowledge transfer across languages.

### 4.1 Settings

We utilized the TREx portion of mLAMA (Kassner et al., 2021) for our experiments. Further information regarding mLAMA and the dataset employed to acquire language vectors can be found in Appendix A.1. We calculated factual knowledge retrieval accuracy as well as English-centric cross-lingual transferability for each language. The details on these evaluation metrics can be found in Appendix A.2. The experiments were based on two multilingual pretrained language models, mBERT[1] and BLOOM[2] (the version with 559 million parameters). The details of probing them can be found in Appendix A.3. Note that the $i$-layer for inserting LIRP and the $j$-layer for inserting LSRP are two hyperparameters, the details on the setting of them can be found in Appendix A.4.

---

[1]https://huggingface.co/bert-base-multilingual-cased
[2]https://huggingface.co/bigscience/bloom-560m

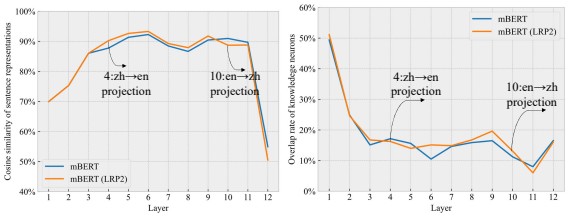

(a) Representation Spaces     (b) Knowledge Neurons

Figure 2: Distance between representation spaces and overlap rate of knowledge neurons in different layers. The languages considered here are Chinese and English. 'mBERT (LRP2)' represents the results with the insertion of the LIRP module into the 4-th layer and the LSRP module into the 10-th layer of mBERT, which yields the best transferability result of Chinese.

| Model | Same | Different | Avg |
|---|---|---|---|
| mBERT | 17.9% | 11.5% | 11.7% |
| mBERT (LRP2) | 18.5% | 11.9% | 12.1% |

Table 2: Overlap rate of knowledge neurons for factual relations in Chinese and English.

## 4.2 Results

Table 1 presents the experimental results on mLAMA, it shows that LRP2 achieves significant improvements in terms of both factual knowledge retrieval accuracy and cross-lingual transferability across various non-English languages over the baseline. The results indicate that cross-lingual alignment of representation spaces indeed facilitates the transfer of rich factual knowledge from English to non-English. More specifically, for both mBERT and BLOOM, LRP2 demonstrates better performance in certain non-Indo-European languages as well as medium- and high-resource languages.

Additional experimental results on X-FACTR (Jiang et al., 2020a) are provided in Appendix B.1. To provide further insights, we present the performance changes for different languages as the number of layers between LIRP and LSRP varies in Appendix B.2. The specific effects of LRP2 on different non-English languages are provided in Appendix B.3. In addition, we observe that the transferability of knowledge shows variations across different types of factual relations, as evidenced in Appendix B.4.

## 5 Working Mechanism of LRP2

In this section, we study the working mechanism of LRP2 from the perspectives of representation space and knowledge neuron.

## 5.1 LRP2 Affects the Alignment of Representation Spaces across Languages

We utilized Chinese-English parallel queries in the mLAMA dataset to collect sentence representations and further calculated the layer-wise cosine similarity of these two languages' sentence representations, as the distance between the representation spaces of these two languages. We conducted a comparative analysis of the distance with and without the utilization of LRP2.

Figure 2a presents the distance between the representation spaces of Chinese and English. It clearly shows the distinct functions of LIRP and LSRP. Specifically, the LIRP module first brings Chinese sentences closer to the representation space of English, thereby facilitating cross-lingual knowledge transfer, while the LSRP module increases the distance between Chinese sentences and the representation space of English, inducing language-specific outputs in Chinese.

## 5.2 LRP2 Enhances the Overlap of Knowledge Neurons across Languages

Dai et al. (2022) discover that knowledge neurons expressing specific factual knowledge exist in pretrained Transformers. Building upon their work, we identify knowledge neurons in multilingual pretrained Transformers and employ them to elucidate the working mechanism of LRP2. The details on how we identify knowledge neurons in multilingual pretrained language models are provided in Appendix C.

Table 2 showcases the overlap rate of knowledge neurons for factual relations in Chinese and English. Notably, we have two interesting findings. First, the overlap rate of knowledge neurons associated with the same relations is considerably higher compared to that with different relations, suggesting the existence of language-independent knowledge neurons within mBERT. Second, LRP2 increases the overlap rate of knowledge neurons between Chinese and English. This improvement indicates that LRP2 facilitates the alignment of English and non-English representation spaces and enhances the activation of knowledge neurons in non-English languages, making them more similar to those in English. In this way, non-English languages acquire factual knowledge transferred from English. Additionally, Figure 2b visualizes the overlap rate of knowledge neurons across different layers. Notably, the layers between LIRP and

LSRP exhibit a prominent increase in the overlap rate of knowledge neurons between Chinese and English.

## 6 Related Work

**Factual Knowledge Probing** Previous works (Petroni et al., 2019; Jiang et al., 2020b) have shown that a language model is a knowledge base. Subsequent works (Kassner et al., 2021; Jiang et al., 2020a) extend monolingual factual knowledge probing to multiple languages. Notably, Jiang et al. (2020a) improve multilingual factual knowledge probing in a code-switching style. Significantly different from this, we suggest that it is essential to allow multilingual pretrained language models to yield language-specific answers.

**Model Editing** A variety of approaches have been proposed to edit knowledge in monolingual language models (Sinitsin et al., 2020; Cao et al., 2021; Mitchell et al., 2022; Meng et al., 2022; Dai et al., 2022). Recently, Xu et al. (2022) define a cross-lingual model editing task, where knowledge updates in one language need to occur in other languages as well. In this paper, we focus on factual knowledge that already exists in multilingual language models and enhance the transferability of them, rather than trying to update a model with new knowledge.

**Cross-lingual Knowledge Transfer** Cross-lingual transfer learning approaches are usually categorized into instance transfer (Zheng et al., 2021; Yang et al., 2022), parameter transfer (Chen et al., 2019; Zhou et al., 2019), and feature transfer (Libovický et al., 2020; Zhao et al., 2021). Most of these works explore cross-lingual knowledge transfer on specific downstream tasks, while we focus on factual knowledge captured by language models and explore the possibility of cross-lingual factual knowledge transfer.

## 7 Conclusion

We have presented a simple yet effective method to transfer factual knowledge from English to non-English languages in multilingual pretrained language models. We empirically confirm that cross-lingual alignment of representation spaces enables factual knowledge transfer across languages in multilingual pretrained language models. Further analysis on knowledge neurons shows that the align-ment of English and non-English representation spaces brought by LRP2 can help non-English languages to stimulate knowledge neurons similar to English, thereby acquiring knowledge transferred from English.

## Limitations

While LRP2 significantly improves factual knowledge retrieval accuracy and facilitates knowledge transferability across diverse non-English languages, it is noteworthy that the LIRP and LSRP modules in LRP2 are inserted into multilingual pretrained language models as two additional layers. Thus, the effectiveness of LRP2 heavily relies on the inherent capabilities of multilingual pretrained language models.

Through extensive experiments conducted on the proposed LRP2 framework, we have demonstrated that cross-lingual alignment of representation spaces enables factual knowledge transfer across different languages. Although this finding is applicable to multilingual pretrained language models of varying architectures, our experiments are limited to two relatively small models due to the limited compute resource available to us. We plan to investigate LRP2 on larger language models when more compute resource is available.

## Acknowledgements

The present research was supported by Zhejiang Lab (No. 2022KH0AB01). We would like to thank the anonymous reviewers for their insightful comments.

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

## A Experiment Details

### A.1 Datasets

mLAMA (Kassner et al., 2021) is a multilingual factual knowledge probing dataset containing 53 languages and 44 factual relations, and the TREx part contains 41 of them. To obtain language vectors, we used OPUS-100 (Zhang et al., 2020) to collect 10,000 filtered sentences for most of the 53 languages, and for languages not included in OPUS-100, such as ceb, we obtained data from the OPUS.[3]

### A.2 Evaluation Metrics

We calculated factual knowledge retrieval accuracy for each language $l$ as $\text{Acc}_l = \frac{|\mathcal{R}_l|}{|\mathcal{D}_l|} * 100$, where $\mathcal{R}_l$ represents the set of correctly predicted knowledge for language $l$ and $\mathcal{D}_l$ represents the entire probing data for language $l$. Additionally, we calculated English-centric cross-lingual transferability as $\text{Trans}_l = \frac{|\mathcal{R}_l \cap \mathcal{R}_{\text{en}}|}{|\mathcal{R}_l \cup \mathcal{R}_{\text{en}}|} * 100$. Here, the denominator $|\mathcal{R}_l \cup \mathcal{R}_{\text{en}}|$ corresponds to the amount of knowledge stored in the probed model, whether in non-English language $l$ or in English form, while the numerator $|\mathcal{R}_l \cap \mathcal{R}_{\text{en}}|$ represents the amount of the stored knowledge both in the form of language $l$ and English, indicating the amount of transferable knowledge.

### A.3 Probing mBERT and BLOOM

Following mLAMA (Kassner et al., 2021), we adopted a typed querying approach for probing. This entails considering all candidate objects of a relation as the candidate pool. For each query associated with a specific relation, we determined the ranking of the correct answer within its candidate pool. The prediction is considered correct if the correct answer is ranked at the top position.

**Probing mBERT**   When probing mBERT, the input query follows the format like "The capital of England is [MASK]", the model's probability predictions for the [MASK] tokens are used to compute the ranking. The number of [MASK] tokens depends on the length of the tokenized object to be predicted. In cases of multiple [MASK] tokens, we calculated the average log probability of these tokens. We utilized the complete candidate pools for probing mBERT (with an average number of candidates per relation of approximately 90).

---

[3] https://opus.nlpl.eu

|  | English (Source) | zh | ko | nl | vi | ceb | ja |
|---|---|---|---|---|---|---|---|
| *Retrieval Accuracy* | | | | | | | |
| mBERT | 22.6 | 14.4 | 12.2 | 18.3 | 22.8 | 14.3 | 10.6 |
| mBERT (LRP2) | 22.6 | **15.4** | **13.3** | **18.8** | **23.3** | **15.6** | **12.8** |
| *English-centric Cross-lingual Transferability* | | | | | | | |
| mBERT | 1 | 30.0 | 24.9 | 48.5 | 46.4 | 25.4 | 23.4 |
| mBERT (LRP2) | 1 | **32.6** | **28.6** | **48.7** | **47.0** | 25.3 | **30.1** |

Table 3: Evaluation results of mBERT on X-FACTR.

**Probing BLOOM**   We notice that the objects to be predicted can appear in the middle of the corresponding query templates in the mLAMA dataset. However, due to the pre-training task of causal language modeling, autoregressive models like BLOOM are more adept at answering factual knowledge questions by predicting the next token in a given query. To address the mismatch between the form of query templates in mLAMA and the generative nature of BLOOM, we employed a compromise approach inspired by Yin et al. (2022). Specifically, when probing the autoregressive BLOOM, we filled each query with objects from its candidate pool to construct complete sentences. We then calculated the model's generation probabilities for these sentences, which serve as the prediction probabilities for different objects. Due to limitation of compute resource, we restricted the size of the candidate pools to 10 when probing BLOOM.

### A.4 Hyperparameters

The $i$-layer for inserting LIRP and the $j$-layer for inserting LSRP are two hyperparameters. We systematically evaluate different combinations of them for each language and report the best results. This exploration allows us to investigate the potential for cross-lingual factual knowledge transfer facilitated by the alignment of representation spaces.

## B Additional Results

### B.1 Experiments on X-FACTR

Yet another dataset used to probe multilingual factual knowledge is X-FACTR (Jiang et al., 2020a). In contrast to mLAMA, this dataset contains fewer languages and slightly more factual relations (23 and 46, respectively). We supplemented experiments on 6 languages of X-FACTR, using mBERT as the baseline model. The results are listed in Table 3, which shows that LRP2 can also achieve improvements on the X-FACTR dataset.

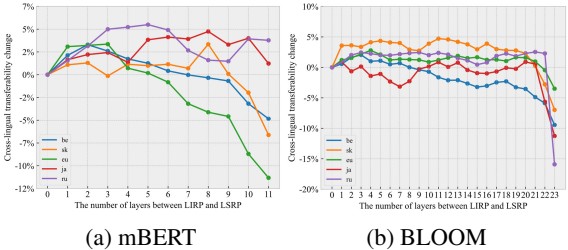

|            |            |
|:----------:|:----------:|
| (a) mBERT  | (b) BLOOM  |

Figure 3: Cross-lingual transferability change with the number of layers between LIRP and LSRP for ru, ja, eu, sk and be. Note that a one-to-many relationship exists between the number of the layers and the configuration of LIRP and LSRP (i.e., the same difference of two numbers corresponds to multiple pairs of numbers). We display the best results for each specific number of layers here.

## B.2 Different Languages Necessitate Varying Optimal Layer Settings

Figure 3 presents the change of cross-lingual transferability for five languages as the number of layers between LIRP and LSRP varies. Notably, we observe that different languages exhibit distinct requirements for representation space alignment to achieve optimal transferability. In addition, we notice that the performance of certain languages is very sensitive to the choice of model layers for the insertion of LIRP and LSRP modules. For certain numbers of layers between LIRP and LSRP for some languages, such as 9 for language eu in Figure 3a, none of the particular insertion settings (the layers where LIRP and LSRP are inserted into are 1/10, 2/11, 3/12, respectively) lead to efficient knowledge transfer. We hypothesize that such sensitivity may stem from the relatively fragile nature of the representation space learned by mBERT for these languages. Consequently, the representations of these languages could easily lose semantic information and become meaningless after language representation projections, leading to a complete failure of knowledge transfer.

In addition, Table 4 and Table 5 show mBERT's and BLOOM's optimal layer configurations for all languages respectively, further underscoring the substantial disparity in the optimal layer settings among various languages.

## B.3 The Impact of LRP2 Differs across Non-English Languages

Figure 4 and Figure 5 illustrate the specific effects of LRP2 on different non-English languages for mBERT and BLOOM respectively. It is note-

worthy that LRP2 is effective for languages that are not covered by the training data of BLOOM. This can be attributed to BLOOM's utilization of a byte-level BPE algorithm for subword tokenization (Scao et al., 2022), ensuring that unknown tokens are never yielded. In this way, unknown languages can be effectively represented to a certain extent, enabling the transfer of factual knowledge between them and other languages.

## B.4 The Transferability Varies across Factual Relations

We assess the transferability change of each factual relation in every language and consider a factual relation to be transferable from English to a non-English language if its transferability improves under any configurations of the LIRP and LSRP modules. Figure 6 illustrates the transferable percentages across all factual relations for mBERT. We observe that 37 out of 41 relations exhibit transferability from English to over 80% non-English languages. Notably, the relations P17, P1412, and P138, representing Place (e.g., Germany, Ireland) and Language (e.g., Italian, Spanish) demonstrate consistent transferability across all languages. However, some factual relations display lower transferability, e.g., P413, P264, P140, and P108, which represent Athlete Position (e.g., midfielder, pitcher), Organization (e.g., Decca, Motown), Religion (e.g., Buddhism, Islam) and Organization (e.g., Apple, Microsoft), respectively.

In addition, Figure 7 reveals a similar trend in the transferability of factual relations between BLOOM and mBERT. Specifically, the factual relations P264, P413 and P449 exhibit lower transferability, while relations representing Place or Language, such as P937, P530, P407, P37, and so on, demonstrate higher transferability in BLOOM.

## C Identifying Knowledge Neurons in Multilingual Pretrained Models

We identify knowledge neurons in multilingual pretrained models using Knowledge Attribution proposed by Dai et al. (2022). We first identify the knowledge neurons of all prompts in a relation. Specifically, for each prompt, we calculate the knowledge attribution scores of neurons and take top-20 neurons as its knowledge neurons. Further, for each factual relation, we take the top-20 neurons with the highest number of occurrences in its all prompts as knowledge neurons

| | ceb | cs | cy | fa | gl | id | ko | lt | pl | pt | ro | sk | ur | vi | af | ar | de | he | hi | ja | zh | es | th | az | bg | bn | da | el | fr | sv | tr | ga | ru | sr | be | ca | eu | hu | hy | it | ka | la | lv | nl | ta | uk | sq | et | fi | ms | hr | sl |
|---|---|---|---|---|---|---|---|---|---|---|---|---|---|---|---|---|---|---|---|---|---|---|---|---|---|---|---|---|---|---|---|---|---|---|---|---|---|---|---|---|---|---|---|---|---|---|---|---|---|---|---|---|
| LIRP | 1 | 1 | 1 | 1 | 1 | 1 | 1 | 1 | 1 | 1 | 1 | 1 | 1 | 1 | 3 | 3 | 3 | 3 | 3 | 3 | 4 | 5 | 5 | 6 | 6 | 6 | 6 | 6 | 6 | 6 | 6 | 7 | 7 | 7 | 8 | 8 | 8 | 8 | 8 | 8 | 8 | 8 | 8 | 8 | 8 | 8 | 9 | 10 | 10 | 10 | 11 | 11 |
| LSRP | 2 | 5 | 2 | 2 | 2 | 2 | 3 | 3 | 2 | 2 | 2 | 9 | 2 | 2 | 4 | 4 | 4 | 6 | 7 | 11 | 10 | 6 | 11 | 7 | 7 | 7 | 7 | 11 | 7 | 7 | 7 | 10 | 12 | 12 | 10 | 9 | 11 | 9 | 9 | 9 | 12 | 9 | 10 | 9 | 9 | 9 | 10 | 11 | 11 | 11 | 12 | 12 |

Table 4: mBERT's optimal layer configurations for all languages. 'LIRP' indicates which layer of mBERT the LIRP module is inserted into, 'LSRP' follows the same pattern.

| | da | ru | sq | ja | ca | es | la | az | cy | af | bg | ceb | et | lt | sl | sr | ta | cs | el | fa | fi | hr | pl | ro | sk | uk | be | hu | hi | hy | ga | id | ka | th | vi | ko | lv | tr | zh | gl | it | nl | eu | pt | fr | de | ms | bn | sv | ur | ar | he |
|---|---|---|---|---|---|---|---|---|---|---|---|---|---|---|---|---|---|---|---|---|---|---|---|---|---|---|---|---|---|---|---|---|---|---|---|---|---|---|---|---|---|---|---|---|---|---|---|---|---|---|---|---|
| LIRP | 1 | 1 | 1 | 2 | 3 | 4 | 4 | 7 | 8 | 8 | 8 | 8 | 8 | 8 | 8 | 8 | 9 | 9 | 9 | 9 | 9 | 9 | 9 | 9 | 9 | 9 | 9 | 10 | 10 | 11 | 11 | 13 | 14 | 15 | 15 | 16 | 16 | 16 | 16 | 17 | 17 | 17 | 17 | 18 | 18 | 18 | 18 | 18 | 20 | 20 | 21 | 21 |
| LSRP | 14 | 22 | 24 | 22 | 22 | 20 | 13 | 9 | 13 | 14 | 22 | 24 | 15 | 13 | 20 | 13 | 19 | 13 | 22 | 22 | 15 | 22 | 20 | 14 | 13 | 21 | 21 | 21 | 16 | 19 | 22 | 19 | 21 | 21 | 22 | 21 | 21 | 23 | 21 | 21 | 22 | 22 | 22 | 23 | 21 | 22 | 23 | 23 | 23 | 22 | 23 | 22 |

Table 5: BLOOM's optimal layer configurations for all languages. 'LIRP' indicates which layer of BLOOM the LIRP module is inserted into, 'LSRP' follows the same pattern.

of it. For a language, we identify knowledge neurons of all its factual relations in mLAMA, such as $\mathcal{KN}_{P101}$, $\mathcal{KN}_{P17}$, etc. Unlike Dai et al. (2022), we perform score ranking at each layer of the model, i.e., for a factual relation, we obtain its knowledge neurons in all layers, e.g., $\mathcal{KN}_{P101} = \{\mathcal{KN}^1_{P101}, \mathcal{KN}^2_{P101}, ..., \mathcal{KN}^L_{P101}\}$, where $L$ is the number of layers of pretrained language models. Specifically, we identified knowledge neurons for both Chinese and English in mBERT. For Chinese, we additionally detected knowledge neurons under the configuration that yields the best transferability result of Chinese.

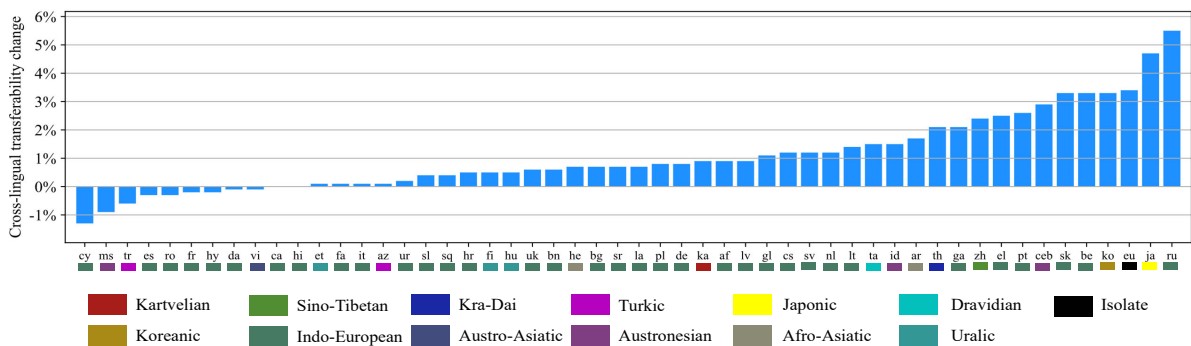

Figure 4: The effect of LRP2 on English-centric cross-lingual transferability of different non-English languages. Results are based on mBERT.

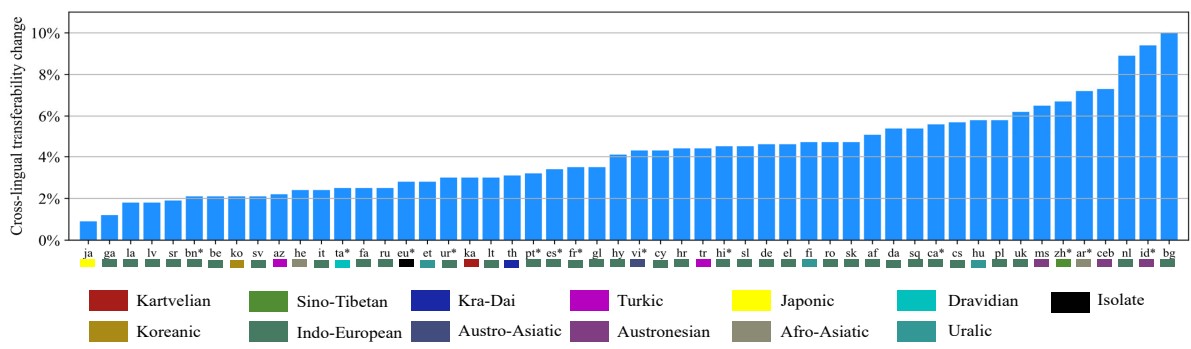

Figure 5: The effect of LRP2 on English-centric cross-lingual transferability of different non-English languages. Results are based on BLOOM. Note that the training data for BLOOM cover only 14 of all languages in the mLAMA dataset, which are marked with an asterisk (*).

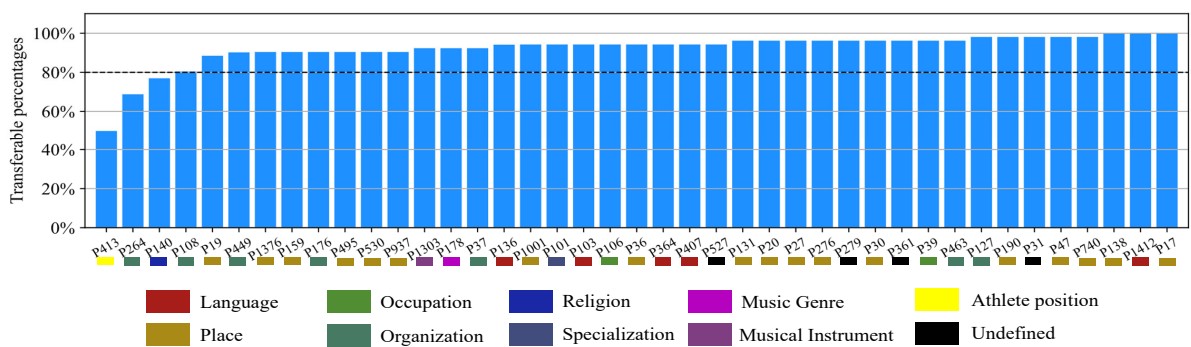

Figure 6: Transferable percentages of all factual relations in mLAMA. Results are based on mBERT.

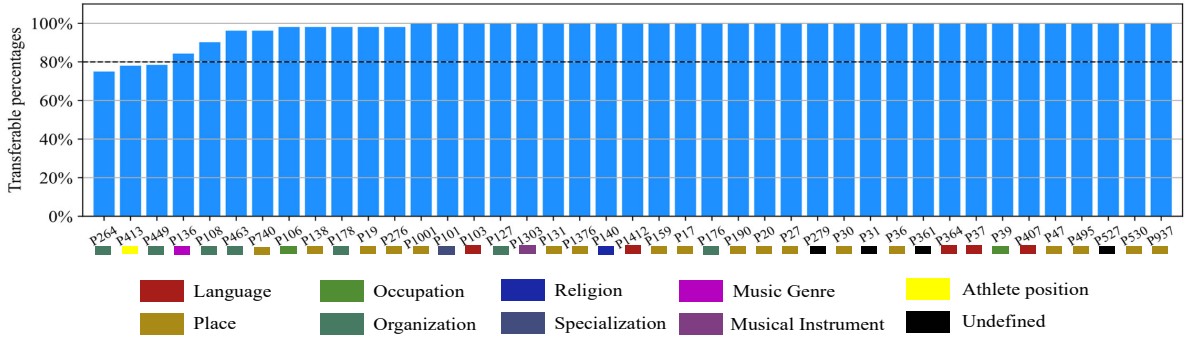

Figure 7: Transferable percentages of all factual relations in mLAMA. Results are based on BLOOM.