# OpenReview forum: "Language Representation Projection: Can We Transfer Factual Knowledge across Languages in Multilingual Language Models?"
_EMNLP/2023/Conference — EMNLP 2023 Main_

### Official Review · Reviewer_LiBy · 2023-08-01

**Typos Grammar Style And Presentation Improvements:** N.A
**Soundness:** 3

**Excitement:**

3: Ambivalent: It has merits (e.g., it reports state-of-the-art results, the idea is nice), but there are key weaknesses (e.g., it describes incremental work), and it can significantly benefit from another round of revision. However, I won't object to accepting it if my co-reviewers champion it.

**Missing References:**

N.A

**Paper Topic And Main Contributions:**

- What is this paper about? This paper aims to transfer factual knowledge from high-resource language to low-resource language
-  what contributions does it make? This paper proposes two parameter-free modules LRP2 to convert non-English representations into English style and then to revert them back. With these proposed modules, this paper demonstrates better results on mLAMA dataset.

**Questions For The Authors:**

N.A

**Reasons To Accept:**

1. This proposes a parameter-free framework LRP2 that enhances the factual knowledge in low resources language in mLAMA dataset.
2. This paper gives promising results for transferring high-resource data to low-resource data by directly projecting the representations in the multilingual pre-trained language models.
3. The parameter-free framework is easy to understand and intuitive.

**Reasons To Reject:**

1. The presentation could be better in this paper. For example, the Figure 2 is not legible.
2. The impacts of the LIRP and LSRP are not given. For example, The attention score/map of some example instances could be given to show the difference between the original attention and projected attention for the understanding of why projecting can help transfer.

**Reproducibility:**

3: Could reproduce the results with some difficulty. The settings of parameters are underspecified or subjectively determined; the training/evaluation data are not widely available.

**Reviewer Confidence:**

4: Quite sure. I tried to check the important points carefully. It's unlikely, though conceivable, that I missed something that should affect my ratings.

---

> ### Author Rebuttal · Authors · 2023-08-29
>
> Thank you for your time to review and your valuable feedback. We respond to your comments below.
>
> 1. The presentation could be better in this paper. For example, the Figure 2 is not legible.
>
> - While the trend depicted in Figure 2 might seem puzzling at first glance, it illustrates the phenomenon where the performance of certain languages, notably "eu" and "sk", is very sensitive to the choice of model layers for the insertion of LIRP and LSRP modules. For certain numbers of layers between LIRP and LSRP for some languages, such as 9 for language "eu" in Figure 2(a), none of the particular insertion settings (the layers where LIRP and LSRP into are inserted are 1/10, 2/11, 3/12, respectively) lead to efficient knowledge transfer. Such sensitivity may stem from the relatively fragile nature of the representation space learned by multilingual pretrained model for these languages. Consequently, the representations of these languages could easily lose semantic information and become meaningless after language representation projections.
> - As this observation is of lesser prominence than the core finding of this paper, we will relocate Figure 2 to Appendix B.5. Additionally, to facilitate reader comprehension of Figure 2, we will provide a more comprehensive analysis of it in Appendix B.5, such as explaining the significant performance fluctuations across different numbers of layers shown in the figure.
> - Furthermore, we will make the following adjustments to enhance the paper's presentation:
>   - Remove details about the proposed method in section 1, and focus the introduction on the background and contributions of this work.
>   - Integrate section 2, "Multilingual Factual Knowledge Probing," into section 1 as a part of the background context.
>   - Present more precise descriptions of method performance and core findings in Section 4.2.
>   - Transfer our insights on knowledge neurons from Appendix C.2 to section 5.
>   - Supplement the analysis of representation space in section 5, including calculations of cosine distance variations between representations of different languages with and without LPR2, as well as 2-D visualizations of sentence representations.
>
> 2. The impacts of the LIRP and LSRP are not given. For example, The attention score/map of some example instances could be given to show the difference between the original attention and projected attention for the understanding of why projecting can help transfer.
>
> - First, the analysis of knowledge neurons in section 5 shows that LRP2 can help non-English languages to stimulate knowledge neurons similar to English, thereby acquiring knowledge transferred from English. This analysis elucidates the underlying mechanism of LRP2 and clarifies the rationale behind the observed knowledge transfer.
>
> - Second, to further investigate the impacts of LIRP and LSRP modules, we have expanded our analysis by utilizing mLAMA to calculate the layer-wise cosine similarity of sentence representations of parallel Chinese and English queries. The results are detailed below, where "mBERT" denotes the direct calculation on mBERT, and "mBERT(LRP2)" represents the statistical outcome with the insertion of the LIRP module into the 4-th layer and the LSRP module the 10-th layer of mBERT, which yields the best transferability result of Chinese.
>
>   |             | 1    | 2    | 3    | 4        | 5    | 6    | 7    | 8    | 9    | 10       | 11   | 12   |
>   | ----------- | ---- | ---- | ---- | -------- | ---- | ---- | ---- | ---- | ---- | -------- | ---- | ---- |
>   | mBERT       | 0.70 | 0.75 | 0.86 | 0.88     | 0.91 | 0.92 | 0.88 | 0.87 | 0.90 | 0.91     | 0.90 | 0.55 |
>   | mBERT(LRP2) | 0.70 | 0.75 | 0.86 | **0.90** | 0.93 | 0.93 | 0.89 | 0.88 | 0.92 | **0.89** | 0.89 | 0.51 |
>
> - This table clearly shows the distinct functions of LIRP and LSRP. Specifically, the LIRP module reduces the distance between representations of different languages, thereby facilitating cross-lingual knowledge transfer, while the LSRP module increases the distance between these representations, inducing language-specific outputs. Furthermore, 2-D visualizations of sentence representations can also help illustrate the role of the two modules.
>
> - We will augment section 5 with the explanation of the impacts of the LIRP and LSRP.
>
> We present a summary of our responses below.
>
> - We will make content adjustments to enhance the paper's presentation. For example, we will relocate Figure 2 to Appendix B.5 and provide a detailed analysis of it there.
> - We will augment section 5 with the explanation of the impacts of the LIRP and LSRP, through the analysis of representation space.

---

### Official Review · Reviewer_fiJ8 · 2023-08-02

**Soundness:** 4

**Excitement:**

3: Ambivalent: It has merits (e.g., it reports state-of-the-art results, the idea is nice), but there are key weaknesses (e.g., it describes incremental work), and it can significantly benefit from another round of revision. However, I won't object to accepting it if my co-reviewers champion it.

**Paper Topic And Main Contributions:**

This paper presents a lightweight method to transfer factual knowledge across languages in multilingually pre-trained LMs. The core assumption of the work is that sentence representations in language X can be projected to the space of language Y by simply adding a fixed language-specific vector v to those representations. The language-specific vectors v_X and v_Y are obtained by averaging over the max-pooled sentence representations of N sentences in language X and Y, respectively.
Given a query in (low-resource) language X, the said vectors are used to push the query representation closer to the representation space of English (based on the assumption that English is the language with most factual knowledge within the model).
More specifically, the addition operation is performed once after layer i to move X representations closer to language space Y, and then after layer j (with j>i) to move them back close to space X.
The method thus does not require a learning phase and is parameter-free.


While this work is promising and addresses a timely and important problem, it has two serious issues:
1) It has been submitted as a short paper, but is really a long paper with a lot of appendices. Much of the appendices' content is essential to understand the paper and support its claims.
This particularly applies to A.2 (evaluation metrics) and C (analysis of knowledge neurons), without which most claims in the conclusions cannot be assessed.
Rearranging the paper to solve this issue can only result in a long submission, in my opinion.

2) The whole method builds on a rather strong assumption that a simple addition of a fixed term to sentence representations in language X can moved them  closer to equivalent representations in language Y. Unfortunately the assumption is not motivated based on previous work, nor is it properly verified empirically (by this, I mean that the paper does not check whether the said addition actually results in representations that are significantly closer to the representations of an equivalent query in the target language).
Instead, the authors present results in terms of accuracy in the knowledge probing task (mLAMA benchmark) and a so-called "English-centric cross-lingual transferability" score, which if I understand correctly *from the appendices* corresponds to the ratio of correct answer overlap between any given language X and English.
The effect of the method on said scores is not convincing: specifically, it is very small for mBERT, and more visible on BLOOM but the latter model has a very low baseline score, which is not compared to the scores reported for similar models in the original mLAMA paper.
Given all this, I find it very hard to judge if the proposed method has a significant effect at all on the problem it set out to solve.

As said above, I do believe this work has potential but it requires more work to be publishable. My recommendations for a resubmission can be summarized as follows:
- make it a long paper with all due explanations in the main body
- probe the direct effect of the vector addition on the query representations (do they  actually get closer, and if so how much, to an equivalent query representation in the target language)
- ensure a proper baseline English score for the BLOOM model (perhaps by improving the probing method currently used on BLOOM, cf. appendix A.3).

**Reasons To Accept:**

The paper presents a lightweight, parameter-free method to improve the accuracy of factual knowledge returned by a multilingual LM in language X, by leveraging the knowledge already existing within the model for the English language.
- the addressed problem is important and timely
- the technique is simple and not computationally costly

**Reasons To Reject:**

Main reasons:
- Some of the core claims of the paper cannot be assessed without reading the appendices (see especially sect. 4.2 and 5). This should really be a long paper.
- The core assumption underlying the proposed method (see above) should be verified or at least motivated on the basis of previous work on cross-lingual representation learning. For instance cf. the strong claim "By performing this projection, the representations of non-English language l are mapped into the English space" in line 164-165.
- The reported improvements are very modest for mBERT. They are better for BLOOM but the baseline scores for this model are very low.

Other issues:
- core equations (1 and 2): can these simple vector additions be called 'projections'?
- The proposed method implies choosing two Transformer layers to which the language-specific vectors are applied. Even though this choice has a crucial effect on the results (Fig 2), it is only vaguely explained in one of the appendices (A.4). It is not entirely clear how much hyper-parameter tuning happened behind the main table of results (Tab 1). This motivates my lower score for reproducibility.
- Fig 2: the 'one-to-many relationship' in this plot makes it hard to understand the presented trends. These plots seem to contain a considerable amount of noise.
- the paper states in different places that the proposed method "significantly" improves factual knowledge accuracy, however said improvements are either very small (for mBERT) or mild on top of a weak baseline (for BLOOM) and I see no mention of statistical significance.

**Reproducibility:**

3: Could reproduce the results with some difficulty. The settings of parameters are underspecified or subjectively determined; the training/evaluation data are not widely available.

**Reviewer Confidence:**

4: Quite sure. I tried to check the important points carefully. It's unlikely, though conceivable, that I missed something that should affect my ratings.

---

> ### Author Rebuttal · Authors · 2023-08-29
>
> We are grateful for your time to review our paper and that you find value in our work. We agree that there are some problems that need to be solved in this work. We respond to your comments below.
>
> 1. Some of the core claims of the paper cannot be assessed without reading the appendices (see especially sect. 4.2 and 5). This should really be a long paper.
>
> - The core content of this paper is straightforward and can be distilled into two key points: 1. we propose a simple method based on language representation projection to explore the feasibility of cross-lingual factual knowledge transfer. 2. we investigate the mechanism behind cross-lingual knowledge transfer from different perspectives. Such clarity makes it easy for us to adjust the structure of the paper.
> - First, we will streamline the paper's content without losing readability as follows:
>   1. Remove details about the proposed method in section 1, and focus the introduction on the background and contributions of this work.
>   2. Integrate section 2, "Multilingual Factual Knowledge Probing," into section 1 as a part of the background context.
>   3. Relocate Figure 2 in the main body and the description of it in section 4.2 to Appendix B.5.
> - Second, we will harness any available space, including an additional fifth page (assuming acceptance), to make the following changes to the main body:
>   1. Present more precise descriptions of method performance and our core findings in Section 4.2.
>   2. Move our insights on knowledge neurons from Appendix C.2 to section 5.
>   3. Supplement the analysis of representation space in section 5, including calculations of cosine distance variations between representations of different languages with and without LPR2, as well as 2-D visualizations of sentence representations.
> - Considering the core content of this paper is clear and simple, we believe it is more suitable for publication as a short paper. To ensure greater clarity and readability, we will make content adjustments that effectively present the paper's central concepts.
>
> 2. The core assumption underlying the proposed method (see above) should be verified or at least motivated on the basis of previous work on cross-lingual representation learning. For instance cf. the strong claim "By performing this projection, the representations of non-English language l are mapped into the English space" in line 164-165.
>
> - First, there are previous studies[1,2] supporting this assumption, and the work we follow to obtain language vectors[1] is one of them, which demonstrates that language-neutral representations can be induced for language X, by subtracting its corresponding language vector. Thereby, there is a certain degree of confidence that another addition of the language Y's language vector to these neutralized representations can futher map language X into the space of language Y. We will explain the motivation behind our method in section 3.
>
> - Second, as suggested by you, we have expanded our analysis by utilizing mLAMA to calculate the layer-wise cosine similarity of sentence representations of parallel Chinese and English queries. The results are detailed in the table below, where "mBERT" denotes the direct calculation on mBERT, and "mBERT(LRP2)" represents the statistical results with the insertion of the LIRP module into the 4-th layer and the LSRP module the 10-th layer of mBERT, which yields the best transferability result of Chinese.
>
>   |             | 1    | 2    | 3    | 4        | 5    | 6    | 7    | 8    | 9    | 10       | 11   | 12   |
>   | ----------- | ---- | ---- | ---- | -------- | ---- | ---- | ---- | ---- | ---- | -------- | ---- | ---- |
>   | mBERT       | 0.70 | 0.75 | 0.86 | 0.88     | 0.91 | 0.92 | 0.88 | 0.87 | 0.90 | 0.91     | 0.90 | 0.55 |
>   | mBERT(LRP2) | 0.70 | 0.75 | 0.86 | **0.90** | 0.93 | 0.93 | 0.89 | 0.88 | 0.92 | **0.89** | 0.89 | 0.51 |
>
> - It illustrates that LRP2 indeed maps the representations of language X to the space of language Y. Furthermore, 2-D visualizations of sentence representations can also confirm this.
>
> - We will augment section 5 with the verification of the assumption that LPR2 can map representations of language X into the space of language Y.
>
> [1] On the Language Neutrality of Pre-trained Multilingual Representations
>
> [2] A Simple and Effective Method To Eliminate the Self Language Bias in Multilingual Representations
>
> 3. The reported improvements are very modest for mBERT. They are better for BLOOM but the baseline scores for this model are very low. & The paper states in different places that the proposed method "significantly" improves factual knowledge accuracy, however said improvements are either very small (for mBERT) or mild on top of a weak baseline (for BLOOM) and I see no mention of statistical significance.
>
> - First, for mBERT, the average English-centric Cross-lingual Transferability without and with LRP2 is 35.2 and 36.3 respectively, indicating that the improvement brought by LRP2 is indeed modest. However, as depicted in Figure 4, LRP2 demonstrates its capacity to yield enhancements across the majority of languages (41 out of 52), showing the noteworthy breadth of its effectiveness. We will provide a more precise description of LRP2's performance.
>
> - Second, we apologize for the oversight during our experimentation on BLOOM. Specifically, in the calculation of sentence generation probabilities for candidate sentences, we mistakenly implemented the calculation as follows: $l_c={1 \over K} \sum_{i=1}^{K}log(p(t_{i-1}|t_{<i}))$, instead of $l_c={1 \over K} \sum_{i=1}^{K}log(p(t_{i}|t_{<i}))$. After rectifying this error, the results became more reasonable (BLOOM's baseline retrieval accuracy on English increased from 12.4 to 35.1). Due to time limitations, we have supplemented experiments on 12 languages of mLAMA. We provide the strong baseline results of BLOOM as follows, where languages marked with an asterisk (*) are covered by BLOOM's pre-training data.
>
>   - Retrieval Accuracy:
>
>   |             | English(source) | zh*      | ja       | eu*      | ga       | vi*      | cy       | es*      | pt*      | ta*      | ur*      | hi*      | ar*      |
>   | ----------- | --------------- | -------- | -------- | -------- | -------- | -------- | -------- | -------- | -------- | -------- | -------- | -------- | -------- |
>   | BLOOM       | 35.1            | 18.7     | 22.7     | 13.3     | 16.4     | 28.9     | 13.7     | 25.9     | 25.8     | 13.2     | 14.4     | 13.5     | 15.1     |
>   | BLOOM(LRP2) | 35.1            | **27.9** | **24.3** | **15.4** | **18.2** | **34.0** | **17.7** | **28.5** | **28.9** | **14.2** | **16.6** | **17.6** | **19.2** |
>
>   - English-centric Cross-lingual Transferability:
>
>   |             | English(source) | zh*      | ja       | eu*      | ga       | vi*      | cy       | es*      | pt*      | ta*      | ur*      | hi*      | ar*      |
>   | ----------- | --------------- | -------- | -------- | -------- | -------- | -------- | -------- | -------- | -------- | -------- | -------- | -------- | -------- |
>   | BLOOM       | 1               | 18.0     | 23.0     | 13.6     | 20.0     | 34.9     | 14.5     | 34.5     | 33.9     | 13.6     | 11.9     | 14.8     | 13.7     |
>   | BLOOM(LRP2) | 1               | **24.7** | **23.9** | **16.4** | **21.2** | **39.2** | **18.8** | **37.9** | **35.9** | **16.1** | **14.2** | **19.3** | **19.2** |
>
> - In addition, we have conducted experiments on another multilingual factual knowledge probing dataset, X-FACTR[1], and also included supplementary experiments on the XLM-RoBERTa-base model[2]. The results of these experiments are presented in the tables below.
>
>   1) mBERT's results on X-FACTR
>
>   - Retrieval Accuracy：
>
>   |             | English(source) | zh       | ko       | nl       | vi       | ceb      | ja       |
>   | ----------- | --------------- | -------- | -------- | -------- | -------- | -------- | -------- |
>   | mBERT       | 22.6            | 14.4     | 12.2     | 18.3     | 22.8     | 14.3     | 10.6     |
>   | mBERT(LRP2) | 22.6            | **15.4** | **13.3** | **18.8** | **23.3** | **15.6** | **12.8** |
>
>   - English-centric Cross-lingual Transferability:
>
>   |             | English(source) | zh       | ko       | nl       | vi       | ceb  | ja       |
>   | ----------- | --------------- | -------- | -------- | -------- | -------- | ---- | -------- |
>   | mBERT       | 1               | 30.0     | 24.9     | 48.5     | 46.4     | 25.4 | 23.4     |
>   | mBERT(LRP2) | 1               | **32.6** | **28.6** | **48.7** | **47.0** | 25.3 | **30.1** |
>
>   2) XLM's results on mLAMA
>
>   - Retrieval Accuracy:
>
>   |           | English(source) | zh       | ga       | cy       | af       | eu   | la      | gl       |
>   | --------- | --------------- | -------- | -------- | -------- | -------- | ---- | ------- | -------- |
>   | XLM       | 24.7            | 15.7     | 9.4      | 10.8     | 14.0     | 9.4  | 6.8     | 18.4     |
>   | XLM(LRP2) | 24.7            | **17.3** | **11.5** | **11.3** | **14.3** | 9.4  | **8.0** | **20.0** |
>
>   - English-centric Cross-lingual Transferability:
>
>   |           | English(source) | zh       | ga       | cy       | af       | eu       | la       | gl       |
>   | --------- | --------------- | -------- | -------- | -------- | -------- | -------- | -------- | -------- |
>   | XLM       | 1               | 28.5     | 17.8     | 20.1     | 33.4     | 15.1     | 12.8     | 36.4     |
>   | XLM(LRP2) | 1               | **31.0** | **22.3** | **21.7** | **36.1** | **15.5** | **17.4** | **37.3** |
>
> - We will revise the new results into the paper to substantiate the method's effectiveness.
>
> [1] X-FACTR: Multilingual Factual Knowledge Retrieval from Pretrained Language Models
>
> [2] Unsupervised Cross-lingual Representation Learning at Scale
>
> 4. Core equations (1 and 2): can these simple vector additions be called 'projections'?
>
> - Thank you for pointing this out. As described in Wikipedia, "In Euclidean geometry, a translation is a geometric transformation that moves every point of a figure, shape, or space by the same distance in a given direction." Since the vector addition used in our method falls within the scope of the "translation" definition, we might adjust the term "projection" involved in our method to "translation" in the next version.
>
> 5. The proposed method implies choosing two Transformer layers to which the language-specific vectors are applied. Even though this choice has a crucial effect on the results (Fig 2), it is only vaguely explained in one of the appendices (A.4). It is not entirely clear how much hyper-parameter tuning happened behind the main table of results (Tab 1). This motivates my lower score for reproducibility.
>
> - As described in Appendix A.4, we conducted a grid search to determine the optimal layer configuration for each language. Certain configuration details are presented below, taking into account the 12 layers of mBERT, numbered from 1 to 12, the term 'LIRP' in the table below indicates which layer of mBERT the LIRP module is inserted into, 'LSRP' follows the same pattern, and 'layer_num' indicates the number of layers between these two modules.
>
> |           | ceb  | cs   | cy   | fa   | gl   | id   | ko   | lt   | pl   | pt   | ro   | sk   | ur   | vi   | af   | ar   | de   | he   | hi   | ja   | zh   | es   | th   | az   | bg   | bn   | da   | el   | fr   | sv   | tr   | ga   | ru   | sr   | be   | ca   | eu   | hu   | hy   | it   | ka   | la   | lv   | nl   | ta   | uk   | sq   | et   | fi   | ms   | hr   | sl   |
> | --------- | ---- | ---- | ---- | ---- | ---- | ---- | ---- | ---- | ---- | ---- | ---- | ---- | ---- | ---- | ---- | ---- | ---- | ---- | ---- | ---- | ---- | ---- | ---- | ---- | ---- | ---- | ---- | ---- | ---- | ---- | ---- | ---- | ---- | ---- | ---- | ---- | ---- | ---- | ---- | ---- | ---- | ---- | ---- | ---- | ---- | ---- | ---- | ---- | ---- | ---- | ---- | ---- |
> | LIRP      | 1    | 1    | 1    | 1    | 1    | 1    | 1    | 1    | 1    | 1    | 1    | 1    | 1    | 1    | 3    | 3    | 3    | 3    | 3    | 3    | 4    | 5    | 5    | 6    | 6    | 6    | 6    | 6    | 6    | 6    | 6    | 7    | 7    | 7    | 8    | 8    | 8    | 8    | 8    | 8    | 8    | 8    | 8    | 8    | 8    | 8    | 9    | 10   | 10   | 10   | 11   | 11   |
> | LSRP      | 2    | 5    | 2    | 2    | 2    | 2    | 3    | 3    | 2    | 2    | 2    | 9    | 2    | 2    | 4    | 4    | 4    | 6    | 7    | 11   | 10   | 6    | 11   | 7    | 7    | 7    | 7    | 11   | 7    | 7    | 7    | 10   | 12   | 12   | 10   | 9    | 11   | 9    | 9    | 9    | 12   | 9    | 10   | 9    | 9    | 9    | 10   | 11   | 11   | 11   | 12   | 12   |
> | layer_num | 1    | 4    | 1    | 1    | 1    | 1    | 2    | 2    | 1    | 1    | 1    | 8    | 1    | 1    | 1    | 1    | 1    | 3    | 4    | 8    | 6    | 1    | 6    | 1    | 1    | 1    | 1    | 5    | 1    | 1    | 1    | 3    | 5    | 5    | 2    | 1    | 3    | 1    | 1    | 1    | 4    | 1    | 2    | 1    | 1    | 1    | 1    | 1    | 1    | 1    | 1    | 1    |
>
> - This table underscores the substantial disparity in optimal layer settings among various languages. Unfortunately, we haven't been able to establish a clear correlation between the specific optimal layer settings (or the number of layers between the two modules) of different languages and their inherent language characteristics, such as language family and language resources, which hinders us from providing precise recommendations for hyperparameter selection.
> - We will include the complete optimal layer settings in Appendix A.4 of the updated version.
>
> 6. Fig 2: the 'one-to-many relationship' in this plot makes it hard to understand the presented trends. These plots seem to contain a considerable amount of noise.
>
> - While the trend in Figure 2 might appear perplexing, it clearly illustrates the phenomenon where the performance of certain languages, notably "eu" and "sk", is very sensitive to the choice of model layers for the insertion of LIRP and LSRP modules. For certain numbers of layers between LIRP and LSRP for some languages, such as 9 for language "eu", none of the particular insertion settings (the layers where LIRP and LSRP are inserted into are 1/10, 2/11, 3/12, respectively) lead to efficient knowledge transfer.
>
> - We believe that removing the "one-to-many relationships" in Figure 2 will make understanding even more difficult. To enhance the readability of the main body, we will move Figure 2 to Appendix B.5 and explain it in more detail, such as explaining the significant performance fluctuations across different numbers of layers shown in the figure.
>
> We present a summary of our responses below.
>
> - We will make content adjustments to ensure greater clarity and readability. For instance, we will relocate Figure 2 from main body to Appendix B.5 and provide a detailed analysis there. Moreover, we will transfer our insights on knowledge neurons from Appendix C.2 to section 5 and supplement the analysis of representation space there.
> - We will explain the motivation behind our method in section 3 and augment section 5 with the verification of the assumption that LPR2 can map representations of language X into the space of language Y.
> - We have corrected the mistake we made in implementing the probing of BLOOM and provided strong baseline results of BLOOM. Furthermore, we have conducted experiments on another multilingual factual knowledge probing dataset, X-FACTR, and included supplementary experiments on the XLM-RoBERTa-base model, to further substantiate the method's effectiveness.
> - We will adjust the term "projection" involved in our method to "translation".
> - We will include the complete optimal layer settings in Appendix A.4.

---

### Official Review · Reviewer_iytb · 2023-08-04

**Soundness:** 4

**Excitement:**

3: Ambivalent: It has merits (e.g., it reports state-of-the-art results, the idea is nice), but there are key weaknesses (e.g., it describes incremental work), and it can significantly benefit from another round of revision. However, I won't object to accepting it if my co-reviewers champion it.

**Paper Topic And Main Contributions:**

This paper investigates the feasibility of transferring factual knowledge from English to non-English languages in multilingual language models. The authors propose a parameter-free framework called Language Representation Projection (LRP2) that aims to improve the transferability of factual knowledge across diverse non-English languages. The results show some performance improvements, but the method relies on strong assumptions, assuming similar distributions of different language spaces. Additionally, the Euclidean distance mapping used in LSRP and LIRP fails to adequately represent the semantic transfer of a sentence between languages. Furthermore, the experiment only utilizes mLAMA, and the exploration of tasks is limited, which raises concerns about the persuasiveness of the findings.

**Reasons To Accept:**

1. The paper presents interesting findings, such as the existence of cross-lingual knowledge neurons in multilingual pre-trained models.

2. The proposed method is simple and demonstrates certain performance improvements.

3. The paper is generally well-written.

**Reasons To Reject:**

1.	The proposed approach should be evaluated through more zero-shot cross-language transfer experiments, extending beyond just mLAMA.
2.	Figure 2 lacks detailed analysis, such as the explanations for the languages (e.g., "eu" and "sk") with significant performance fluctuations across layers.
3.	The paper does not provide a recommendation for selecting the number of layers between LIRP and LSRP, which is a critical hyperparameter affecting performance across different languages and tasks. This lack of guidance limits the application and further extension of the proposed method.

**Reproducibility:**

3: Could reproduce the results with some difficulty. The settings of parameters are underspecified or subjectively determined; the training/evaluation data are not widely available.

**Reviewer Confidence:**

2: Willing to defend my evaluation, but it is fairly likely that I missed some details, didn't understand some central points, or can't be sure about the novelty of the work.

---

> ### Author Rebuttal · Authors · 2023-08-29
>
> We thank you for your valuable feedback and your interest in our methodology and findings. We present our responses to your comments below.
>
> 1. The proposed approach should be evaluated through more zero-shot cross-language transfer experiments, extending beyond just mLAMA.
>
> - Another dataset used to probe multilingual factual knowledge is X-FACTR[1]. In contrast to mLAMA, this dataset contains fewer languages and slightly more factual relations - 23 and 46, respectively. Due to time limitations, we have supplemented experiments on 6 languages of X-FACTR, using mBERT as the baseline model.  The results are listed below：
>
>   - Retrieval Accuracy：
>
>   |             | English(source) | zh       | ko       | nl       | vi   | ceb      | ja       |
>   | ----------- | --------------- | -------- | -------- | -------- | ---- | -------- | -------- |
>   | mBERT       | 22.6            | 14.4     | 12.2     | 18.3     | 22.8 | 14.3     | 10.6     |
>   | mBERT(LRP2) | 22.6            | **15.4** | **13.3** | **18.8** | 23.3 | **15.6** | **12.8** |
>
>   - English-centric Cross-lingual Transferability：
>
>   |             | English(source) | zh       | ko       | nl       | vi       | ceb  | ja       |
>   | ----------- | --------------- | -------- | -------- | -------- | -------- | ---- | -------- |
>   | mBERT       | 1               | 30.0     | 24.9     | 48.5     | 46.4     | 25.4 | 23.4     |
>   | mBERT(LRP2) | 1               | **32.6** | **28.6** | **48.7** | **47.0** | 25.3 | **30.1** |
>
> - It shows that LRP2 can also achieve improvements on X-FACTR dataset. The results of the  X-FACTR experiment will be included in Appendix B.4 of the updated version.
>
> [1] X-FACTR: Multilingual Factual Knowledge Retrieval from Pretrained Language Models
>
> 2. Figure 2 lacks detailed analysis, such as the explanations for the languages (e.g., "eu" and "sk") with significant performance fluctuations across layers.
>
> - Previous works[1,2] find that different layers within multilingual pretrained models have very different effects on cross-lingual transfer. Through Figure 2, we aim to provide further insight into the phenomenon that, even within the same multilingual pretrained model, different languages necessitate varying numbers of model layers to achieve optimal cross-lingual knowledge transfer.
> - The phenomenon of "significant performance fluctuations across layers", notably witnessed in certain languages (e.g. "eu" and "sk"), illustrates that the performance of these languages is very sensitive to the choice of model layers to insert LIRP and LSRP modules. For certain numbers of layers between LIRP and LSRP for some languages, such as 9 for language "eu" in Figure 2(a), none of the particular insertion settings (the layers where LIRP and LSRP are inserted into are 1/10, 2/11, 3/12,  respectively) lead to efficient knowledge transfer.
> - This sensitivity may stem from the fact that the representation space learned by the multilingual pretrained model for these languages is relatively vulnerable, and only representation projections at specific locations can preserve the meaningfulness of the representations, thus enabling successful knowledge transfer. Conversely, in other settings, the representations of these languages easily lose semantic information and become meaningless, leading to a complete failure of knowledge transfer.
> - We will make the following changes to the content of the paper：
>   - To enhance the readability of the main body, we will relocate Figure 2 and a detailed analysis of it into Appendix B.5.
>   - We will delve deeper into our findings regarding the central question explored in this paper in section 4.2, and briefly mention the phenomenon of "significant performance fluctuations across layers" introduced by LRP2 there.
>
> [1] INFOXLM: An Information-Theoretic Framework for Cross-Lingual Language Model Pre-Training
>
> [2] Investigating Language Relationships in Multilingual Sentence Encoders Through the Lens of Linguistic Typology
>
>
>
> 3. The paper does not provide a recommendation for selecting the number of layers between LIRP and LSRP, which is a critical hyperparameter affecting performance across different languages and tasks. This lack of guidance limits the application and further extension of the proposed method.
>
> - As described in Appendix A.4, we conducted a grid search to determine the optimal layer configuration for each language. Certain configuration details are presented below, taking into account the 12 layers of mBERT, numbered from 1 to 12, the term 'LIRP' in the table below indicates which layer of mBERT the LIRP module is inserted into, 'LSRP' follows the same pattern, and 'layer_num' indicates the number of layers between these two modules.
>
>   - mBERT's optimal layer configuration on mLAMA
>
>   |           | zh   | ko   | nl   | vi   | ceb  | ja   |
>   | --------- | ---- | ---- | ---- | ---- | ---- | ---- |
>   | LIRP      | 4    | 1    | 8    | 1    | 1    | 3    |
>   | LSRP      | 10   | 3    | 9    | 2    | 2    | 11   |
>   | layer_num | 6    | 2    | 1    | 1    | 1    | 8    |
>
>   - mBERT's optimal layer configuration on X-FACTR
>
>   |           | zh   | ko   | nl   | vi   | ceb  | ja   |
>   | --------- | ---- | ---- | ---- | ---- | ---- | ---- |
>   | LIRP      | 8    | 8    | 4    | 1    | 3    | 3    |
>   | LSRP      | 11   | 12   | 5    | 2    | 4    | 11   |
>   | layer_num | 3    | 4    | 1    | 1    | 1    | 8    |
>
> - First, both tables above underscore the substantial disparity in the optimal layer settings among various languages. Unfortunately, we haven't been able to establish a clear correlation between the specific optimal layer settings (or the number of layers between the two modules) of different languages and their inherent language characteristics, such as language family and language resources, which hinders us from providing precise recommendations for hyperparameter selection.
>
> - Second, we observe that for some languages, like vi and ja, mBERT's optimal layer settings remain consistent across different datasets. This indicates that for certain languages, the identified hyperparameter are dataset-independent and could potentially be applied across diverse tasks to facilitate cross-lingual knowledge transfer.
>
> - We will include the optimal layer settings and our findings in Appendix A.4 of the updated version.
>
>
>
> 4. The method relies on strong assumptions, assuming similar distributions of different language spaces. & The Euclidean distance mapping used in LSRP and LIRP fails to adequately represent the semantic transfer of a sentence between languages.
>
> - The core assumption behind our method is that through a Euclidean distance mapping, the representation space between language A and language B can be transferred. This kind of simple mapping is certainly relatively coarse and cannot achieve precise semantic transfer. Perhaps the over-idealization of the method's working mechanism in Figure 1 has led to misunderstandings. We will provide a precise clarification of our core assumption in the description section of Figure 1.
>
> - To explore the underlying mechanism of LRP2, we conducted an analysis of multilingual knowledge neurons in section 5. The analysis results demonstrate that LRP2 can enhance the overlap of knowledge neurons across languages, revealing the actual working mechanism of LRP2.
>
> - Furthermore, we have expanded our analysis by utilizing mLAMA to calculate the layer-wise cosine similarity of sentence representations of parallel Chinese and English queries. The results are detailed below, where "mBERT" denotes the direct calculation on mBERT, and "mBERT(LRP2)" represents the statistical outcome with the insertion of the LIRP module into the 4-th layer and the LSRP module the 10-th layer of mBERT.
>
>   |             | 1    | 2    | 3    | 4        | 5    | 6    | 7    | 8    | 9    | 10       | 11   | 12   |
>   | ----------- | ---- | ---- | ---- | -------- | ---- | ---- | ---- | ---- | ---- | -------- | ---- | ---- |
>   | mBERT       | 0.70 | 0.75 | 0.86 | 0.88     | 0.91 | 0.92 | 0.88 | 0.87 | 0.90 | 0.91     | 0.90 | 0.55 |
>   | mBERT(LRP2) | 0.70 | 0.75 | 0.86 | **0.90** | 0.93 | 0.93 | 0.89 | 0.88 | 0.92 | **0.89** | 0.89 | 0.51 |
>
> - The above results illustrate that LRP2 first diminishes the disparity between semantic representations of different languages and then increases it, providing another explanation for the mechanism behind LRP2. We will include the analysis of representation space of different languages in section 5.
>
> We present a summary of our responses below.
>
> - We have supplemented experiments on another multilingual factual knowledge probing dataset to further demonstrate the effectiveness of the method.
> - We will move Figure 2 to Appendix B.5 and analyze it in more detail, such as explaining the phenomenon of "significant performance fluctuations across layers"
> - We will include the optimal layer settings and our findings in Appendix A.4 of the updated version.
> - We will add an analysis of the similarities between representations of different languages to further explore the underlying mechanism of LRP2 in section 5.

---

### Meta-Review · Area_Chair_FoR1 · 2023-09-19

**Recommendation:** 4

**Metareview:**

This paper proposes and explores a lightweight approach to transfer factual knowledge across languages in multilingually pre-trained language models by using Language Representation Projection modules (LRP2). Overall, the paper presents a consistent takeaway that usage of LRP2 on multiple multilingual models shows improvement in the knowledge probing tasks using the mLAMA dataset. However, as pointed out by multiple reviewers, many important details are packed into the appendix and also into the discussion posts. The AC strongly recommends including it in the appendix and on the fifth page (in case of acceptance) given that the paper is pretty dense.

---

### Decision · Program_Chairs · 2023-10-07

**Decision:**

Accept-Main

**Comment:**

This paper proposes and explores a lightweight approach to transfer factual knowledge across languages in multilingually pre-trained language models by using Language Representation Projection modules (LRP2). Overall, the paper presents a consistent takeaway that usage of LRP2 on multiple multilingual models shows improvement in the knowledge probing tasks using the mLAMA dataset. However, as pointed out by multiple reviewers, many important details are packed into the appendix and also into the discussion posts. The AC strongly recommends including it in the appendix and on the fifth page (in case of acceptance) given that the paper is pretty dense.